# MaskBEV: Towards A Unified Framework for BEV Detection and Map Segmentation

## ABSTRACT

Accurate and robust multimodal multi-task perception is crucial for modern autonomous driving systems. However, current multimodal perception research follows independent paradigms designed for specific perception tasks, leading to a lack of complementary learning among tasks and decreased performance in multi-task learning (MTL) due to joint training. In this paper, we propose MaskBEV, a masked attention-based MTL paradigm that unifies 3D object detection and bird's eye view (BEV) map segmentation. MaskBEV introduces a task-agnostic Transformer decoder to process these diverse tasks, enabling MTL to be completed in a unified decoder without requiring additional design of specific task heads. To fully exploit the complementary information between BEV map segmentation and 3D object detection tasks in BEV space, we propose spatial modulation and scene-level context aggregation strategies. These strategies consider the inherent dependencies between BEV segmentation and 3D detection, naturally boosting MTL performance. Extensive experiments on nuScenes dataset show that compared with previous state-of-the-art MTL methods, MaskBEV achieves 1.3 NDS improvement in 3D object detection and 2.7 mIoU improvement in BEV map segmentation, while also demonstrating slightly leading inference speed.

## CCS CONCEPTS

• **Computing methodologies → Scene understanding**; **Vision for robotics**.

## KEYWORDS

3D perception, multi-task learning, bird's eye view, BEV map segmentation

## 1 INTRODUCTION

Perceiving the 3D environment around a vehicle is crucial for autonomous driving systems. Lidar and cameras are widely used in autonomous driving fusion perception due to their complementary characteristics. Some object-centric methods [1, 7, 38, 40, 41, 52] have carefully designed multimodal fusion perception modules to enhance the performance of 3D object detection. However, they are difficult to adapt to multi-task requirements and lack flexibility in generalizing to other tasks. These shortcomings limit their practical

Unpublished working draft. Not for distribution.

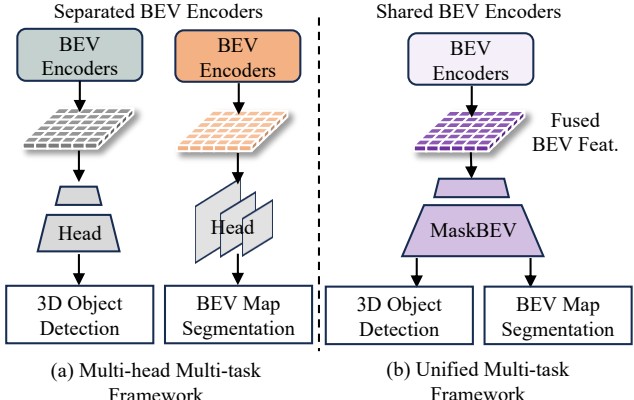

**Figure 1: Comparison between the multi-head multi-task perception framework with separated BEV encoder and our proposed MaskBEV. (a) Multiple task heads implement multi-task learning (MTL). The previous methods [11, 28, 35] adopt independent task head design. (b) The unified multi-task head design fully exploits the complementary advantages between multiple tasks, and uses one decoder to perform MTL in the unified BEV features.**

application. The traditional single-task perception paradigm is gradually shifting towards multi-task learning (MTL), such as sparse 3D detection tasks and dense BEV map segmentation tasks. Based on dense bird's eye view (BEV) representations, a feasible solution is provided, which has received widespread attention due to its natural support for multi-task perception. However, experiments by [11, 28] have found that current MTL paradigms are affected by the negative transfer problem of multitasking.

BEVFusion [28] proposed that joint training with a shared BEV encoder led to a decrease in MTL performance, and then mitigated the negative transfer of MTL by separating the BEV encoder during training, as shown in Fig.1(a). MetaBEV [11] adopted the routing multi-task mixture-of-experts technology of natural language processing (NLP) and separated BEV features to improve MTL, but its MTL accuracy is still much lower than that of single tasks. The powerful UniTR unified the image and LiDAR encoder backbones, but more importantly, these state-of-the-art (SOTA) works [11, 28, 35] still employed independent prediction head designs, such as the Transformer head for 3D detection [1, 36] and the CNN head [28, 49] for map segmentation, as shown in Fig.1(a). Then, MTL is achieved through a simple combination of 3D detection and BEV segmentation task heads. The design of these multitask methods leads to unnecessary increases in computational costs and performance degradation, with complementary features between tasks not being

utilized. In this paper, we aim to extend the current multimodal fusion framework by designing a multi-task complementary learning decoder to construct a unified multi-task perception framework.

In this paper, we introduce MaskBEV, a unified multi-task outdoor 3D perception framework. As shown in Fig. 1(b), unlike previous task-specific perception heads, our MaskBEV is the first to achieve simultaneous perception of 3D object detection and BEV map segmentation in one decoder head. To achieve this aim, we adopt the advanced Mask2Former [9] paradigm, leveraging the complementary nature of the BEV map segmentation task and the 3D object detection task to construct a unified multi-task decoder head. Masked attention focuses attention on local features centered around potential queries. We utilize the union of multi-task masks in BEV space to guide query-based feature learning. To maximize the coverage of masks over potential regions of interest while excluding the entire BEV space, we introduce a spatial modulation strategy that fully considers the geometric relationships of detection and the semantic principles of segmentation. Moreover, we propose a powerful scene-level feature aggregation module to aggregate multi-granular contextual features to serve the BEV map segmentation task better. Specifically, the module consists of two BEV feature aggregation blocks. The multi-window window-attention (MWWA) adjusts window sizes on different attention heads to aggregate multi-granularity contextual features. ASPP [5] implements scene-level global feature extraction from BEV feature maps in a convolution-based manner. Structurally, Transformer-based MWWA and convolution-optimized ASPP are complementary, and the performance gain also demonstrates the effectiveness of this module.

The query-based decoding paradigm naturally fits the current 3D object detection, the mask decoder structure achieves the segmentation of the BEV map, and the query focus on foreground regions allows for better updating of queries. In summary, our main contributions are as follows:

- We propose MaskBEV which is a unified perception framework for 3D object detection and BEV map segmentation tasks for the first time. The proposed multi-task decoder based on masked attention can achieve high-performance joint training.
- We propose a spatial modulation strategy to assist in obtaining multi-task reliable masks and a new scene-level feature aggregation module to capture multi-granularity and even scene-level BEV contextual features.
- Our MaskBEV achieves state-of-the-art performance on multitask learning (3D object detection and BEV map segmentation) on nuScenes dataset. Multiple multimodal feature encoder networks and sensor robustness analyses are also provided for a comprehensive evaluation of MaskBEV.

## 2 RELATED WORK

### 2.1 3D Object Detection

3D object detection is one of the key tasks in autonomous driving perception. The performance of Lidar-only methods [18, 39, 42] and camera-only methods [14, 15, 22, 37] is limited by the deficiencies of their respective sensors. Multimodal fusion methods [1, 6, 8, 11, 21, 28, 35, 38] recently show significant effectiveness in 3D

object detection. Object-centric detection methods [7, 16, 38, 41, 52] improve detection accuracy by carefully designing potential object query proposal generation and generation modules. CMT [38] enriches multimodal 3D features with coordinate encoding and designs a 3D object detection decoder head through the original Transformer decoder in DETR. SparseFusion [52] extracts sparse instance features from the multimodal and fuses them directly to obtain the final sparse instance features for detection. In addition, some methods [11, 17, 23, 28, 35] mainly use BEV representation to fuse the two modalities. BEVFusion [28] applies lift-splat-shoot (LSS) [31] operations to project image features onto BEV space and fuses Lidar BEV features in that space. Then, the improved TransFusion [1] decoder head is used for 3D detection. Current SOTA methods [11, 17, 35] focus on generating BEV features and applying this detection head. UniTR [35] achieves unified feature encoding of images and Lidar through the modality-independent Transformer encoder. It is worth noting that object-centric method [7, 38, 40, 41] is difficult to extend to the BEV map segmentation task.

### 2.2 BEV Map Segmentation

BEV map segmentation is the task of performing dense semantic segmentation in a bird's eye view. Influenced by the development of 3D object detection in BEV representation, BEV map segmentation [22, 31, 42] has recently received considerable attention. Such as LSS [31] achieves BEV map segmentation through ResNet-18 [13] and a multi-scale feature fusion network. Some works [20, 37] transform images into BEV views through feature projection. BEV-Former [22], CVT [49], BEVSegFormer [30], and MetaBEV [11] construct BEV representations in a learnable manner[54] and they adopt a convolution-based segmentation head similar to the head of LSS. Convolution-based segmentation heads [31, 49] are widely used in current SOTA BEV map segmentation methods [11, 28, 35]. Additionally, PETR V2 [26] proposes a query-based segmentation head from the vanilla DETR [4]. These perception methods aim to transform each sensor feature into BEV space to achieve multi-task prediction, including BEV segmentation.

### 2.3 Multi-Task Learning

MTL has garnered widespread attention and mutual reinforcement in both computer vision and NLP fields. Previous multi-task research can be roughly divided into camera-only methods [22, 26, 37], Lidar-only methods [18, 42], and cross-modal fusion methods [11, 23, 28, 33, 35, 43]. Camera-only methods mostly convert multi-view cameras into BEV feature maps, and perform 3D object detection or BEV segmentation based on BEV map combined with specific task heads. Lidar-only methods extract features through the point cloud network [18, 51] and compress them in the $Z$-axis direction to obtain BEV representation. Some general MTL works [11, 22, 28, 35, 42, 43, 48] design unified BEV representations to achieve multi-task perception, including sparse detection tasks and dense semantic tasks. However, due to the adoption of independent task heads in each architecture, MTL performance is adversely affected by task conflicts [28], resulting in poorer performance. Some works [11, 28] adopt separating BEV encoders to mitigate the negative transfer of multi-task joint training on each single task. In

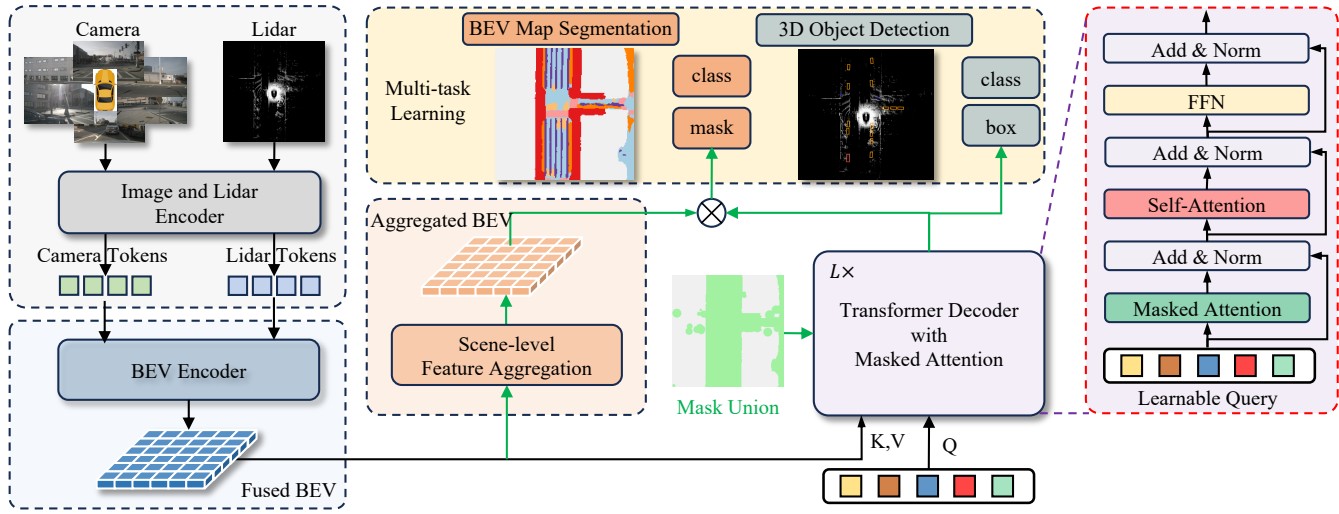

**Figure 2: Overview of our MaskBEV framework. Multimodal input is passed through the feature encoding network to obtain the fused BEV features. Based on unified BEV features, our MaskBEV performs BEV map segmentation and 3D detection tasks on a unified Transformer decoder. Multi-task perception is not a simple task stacking, but a composite task learning process that promotes each other by utilizing the complementary characteristics of tasks.**

contrast to all existing methods, MaskBEV does not independently design task-specific decoder heads. Instead, it naturally integrates the individual characteristics of BEV segmentation and object detection tasks, mutually reinforcing the two tasks in a shared decoder head. MaskBEV represents a new Transformer-based paradigm for MTL based on unified BEV representations.

## 3 METHODS

### 3.1 Overall Architecture

In this paper, we introduce a new unified multi-task learning decoder to address the performance degradation issues of 3D object detection and BEV map segmentation in joint training. Fig. 2 illustrates the architecture of MaskBEV. Given multimodal inputs, they are encoded into tokens using a multimodal feature encoder, and then fused into a BEV space through a BEV encoder [28]. Finally, a decoder based on advanced Mask2Former [9] is used to perform various 3D perception tasks. Our main innovation focuses on the decoder module. Convert the multi-task perception results into a binary mask in masked attention, allowing the query to focus on the local region of the entire BEV map (Section 3.3). The decoder decodes segmentation predictions as Transformer-based mask classification and detection predictions into basic classification and regression. Scene-level feature aggregation fuses multi-scale features to facilitate the BEV map segmentation task (Section 3.4).

### 3.2 Lidar-Camera Feature Encoder

BEV features can be sourced from most SOTA feature encoding backbone networks [11, 28, 35]. In our research, we take UniTR [35] as an example to process multimodal inputs to generate BEV features. Specifically, modality-specific tokenizers [10, 50] process

multimodal signals to generate input token sequences for subsequent Transformer encoders. Image and Lidar tokens learn complementary features through modality-agnostic Transformer blocks based on DSVT blocks [34]. Camera and Lidar feature tokens are fused into a unified BEV space through a convolution-based BEV encoder [28]. We use the function $f(\cdot)$ to represent the multimodal feature encoding process:

$$F = f(F_C, F_L), \qquad (1)$$

where $F$ is the BEV feature, $F \in \mathbb{R}^{C \times H \times W}$, $C$ is the channel dimension, $H$ and $W$ are the BEV feature map sizes. $F_C$ is the camera feature, and $F_L$ is the Lidar feature. The BEV features are fed to the decoder for multi-task predictions.

### 3.3 Unified Multi-Task Transformer Decoder

Previous SOTA works [11, 28, 35] adopt a Transformer-based 3D detection head [1] and a CNN-based segmentation head [49] to implement MTL via a simple union as shown in Fig. 1(a). However, these methods often rely on task-specific decoders and do not consider the unified modeling and complementary effects of multiple tasks, which can enhance the performance of any single task. To this end, inspired by the advanced Mask2Former [9] decoder design, we propose the unified MTL framework MaskBEV. As shown in Fig. 2, the region of interest for MaskBEV is only a small part of the entire BEV map. We attempt to focus the cross-attention between object queries and the BEV feature on the masks of potential tasks rather than focusing on the whole BEV. The predicted results can explicitly guide the update of query features. The self-attention between object queries in the decoder infers the pairwise relationships between different queries.

Specifically, with the input BEV features $F \in \mathbb{R}^{C \times H \times W}$ and a set of parameterized query features $Q \in \mathbb{R}^{C \times N}$, $N$ is the query number.

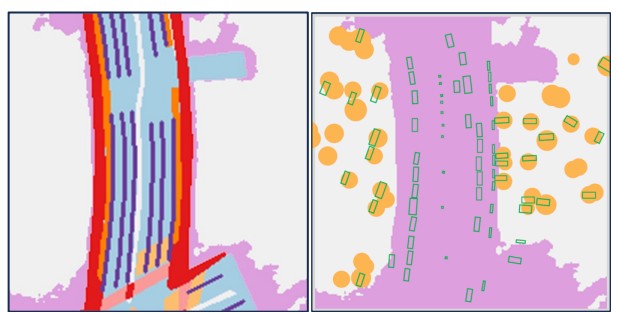

Figure 3: Illustration of attention mask. Left, purple represents the modulated mask, and we superimpose the ground truth. Right, yellow represents modulated 3D objects and green boxes represent ground truth. The mask of an object whose center point is on the segmentation mask is not drawn.

Define the anchor for each query $A_q = (x_q, y_q, z_q, l_q, w_q, h_q, \theta_q)$, where $x_q, y_q$ is the center point, $z_q$ is bounding box height, $l_q, w_q$, and $h_q$ is length, width and height, $\theta_q$ is yaw angle. We encode the anchor of each query through a multi-layer perception (MLP) to obtain the position embedding $P_q$:

$$P_q = \text{MLP}(\text{PE}(A_q)) = \text{MLP}(\text{Cat}(\text{PE}(x_q), ..., \text{PE}(\theta_q))), \quad (2)$$

where $\text{PE}(A_q) \in \mathbb{R}^{2C}$, MLP implements $R^{2C} \rightarrow R^C$, $P_q \in \mathbb{R}^C$, Cat means concatenate function.

The query consists of position encoding $P_q$ and learnable content query $C_q$, $Q = P_q + C_q$ [4, 25]. This allows the network to learn context and location features simultaneously.

The transformer decoder performs an iterative updating of the query features toward the desired 3D object detection and BEV map segmentation. Specifically, in each iteration layer $l$, the query $Q_l$ focus on their corresponding regions through masked attention:

$$Q_{l+1} = \text{Softmax}(\mathcal{M}_{l-1} + Q_l K_l^T) V_l + Q_l, \quad (3)$$

where $K_l, V_l = FW_k, FW_v$, $W_k$ and $W_v$ are parameters of linear projection. $\mathcal{M}_{l-1}$ is the multi-task union mask of the attention mask from the previous layer.

Specifically, the attention mask is the union of 3D objects in BEV and BEV map segmentation. However, during the training process, predictions of potential objects and regions are inaccurate, and predicted boxes cannot effectively represent the precise location of objects in BEV space. To address this, we propose a spatial modulation strategy to ensure that the attention mask covers as many objects and semantic regions as possible. Firstly, we use BEV segmentation prediction results with a threshold greater than 0.1 as the segmentation masks. For 3D objects, we use the top 200 box prediction results since the max number of objects in one frame is 142, and draw circular regions of interest with the predicted center point of the box as the center and 1.3 times the length of the box as the diameter to create object masks. Fig. 3 shows a visual example of attention masks.

After each iteration, on the one hand, the feed-forward network (FFN) independently decodes $N$ object queries containing instance information into 3D boxes and class labels. The FFN predicts $\delta x, \delta y, z, log(l), log(w), log(h), sin(\theta), cos(\theta)$ of 3D anchor box. And

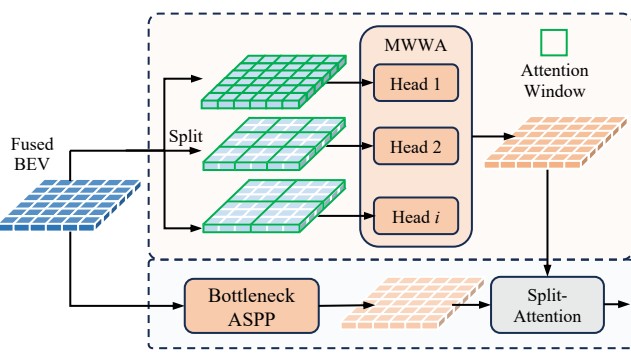

Figure 4: Illustration of scene-level feature aggregation. In MWWA, multi-attention heads independently calculate attention in windows of different sizes to capture multi-scale features. ASPP captures the scene-level semantic layout of BEV features.

predict the per class probability ($p \in [0, 1]^K$) of $K$ object semantic classes. More details of FNN are consistent with previous 3D detection paper [1]. On the other hand, each query $q_i$ is projected to predict its semantic logits $S_i$ and the mask embedding $E_{mask,i}$, $E_{mask,i} \in \mathbb{R}^C$. Then do the dot product of $E_{mask,i}$ with the BEV features, $F_a \in \mathbb{R}^{C \times H \times W}$, aggregated by scene-level feature aggregation (see Section 3.4). Finally, the binary BEV mask is obtained through a sigmoid function.

$$Mask_{b,i} = \delta(E_{mask,i} \odot F_a), \quad (4)$$

where $\delta(\cdot)$ is a sigmoid function, $Mask_{b,i} \in \mathbb{R}^{H \times W}$. BEV map semantic segmentation prediction results $Mask_s$ are as follows:

$$Mask_s = \sum_{i=1}^{N} S_i \cdot Mask_{b,i}. \quad (5)$$

## 3.4 Scene-Level Feature Aggregation

We propose scene-level feature aggregation to capture multi-granular contextual BEV features. Inspired by recent progress in introducing windows into Transformer [19, 27, 47], we design scene-level feature aggregation as a hybrid structure, as shown in Fig. 4. For the input fused BEV features, multi-window windowed attention (MWWA) performs window attention with windows of different sizes. MWWA aggregates multi-granular contextual semantic features on each attention head. In addition, we apply ASPP [5] and the bottleneck structure [12] to reduce the channel number by 4× to capture the global context. Finally, the dual outputs are weighted and fused through lightweight split-attention [46] and skip-connected with the original BEV feature map.

Specifically, as shown in Fig. 4(a), MWWA uses a pyramid window setting. Each window has a different attention range to capture multi-granular contextual information. It is worth noting that, our MWWA has different scale windows for each head, unlike standard multi-head window attention. Specifically, set the window size of $i$-th head $h_i$ to $P_i \times P_i$. Given the input BEV features $F \in \mathbb{R}^{C \times H \times W}$, and split it into $M$ sub-BEV features, $F_i \in \mathbb{R}^{D \times H \times W}$, $1, 2, \cdots, M$,

where $D$ is the sub-BEV feature dimension. We assign each sub-BEV feature an attention head for a specific window $P_i \times P_i$. To dynamically focus on the remote region, we also use the shifted window operation following [27]. MWWA can be expressed as:

$$\text{MWWA}(F_i, P_i) = \text{Cat}(h_1, h_2, \cdots, h_l), \quad (6)$$

$$h_i = f_{\text{WA}}(F_i, P_i), \quad (7)$$

where $l$ is the number of heads, Cat means concatenate function, $f_{\text{WA}}(\cdot)$ is the window attention.

## 4 EXPERIMENTS

### 4.1 Dataset

We evaluate the performance of MaskBEV by comparing it with exiting SOTA methods on nuScenes [3] dataset. nuScenes is a very challenging large-scale autonomous driving dataset, containing 700 scenes for training, 150 scenes for validation, and 150 scenes for testing. It provides point clouds collected with 32-beam Lidar and six cameras with complete 360 environment coverage. The annotated data can be widely used in tasks such as 3D object detection, object tracking, and BEV map segmentation. Following [28, 35], we set the detection range to $[-51.2m, 51.2m]$ for the $X$ and $Y$ axes, and $[-5m, 3m]$ for the $Z$ axis and the segmentation range to $[-50m, 50m]$ for the $X$ and $Y$ axes. Our evaluation metrics align with [3, 28]. For 3D detection, we utilize the standard nuScenes detection score (NDS) and mean average precision (mAP). For BEV map segmentation, we follow [28, 49] to calculate the mean intersection over union (mIoU) on the overall six categories (drivable space, pedestrian crossing, walkway, stop line, car-parking area, and lane divider). The input camera and Lidar size depend on the specific BEV encoding backbone network.

### 4.2 Implementation Details

**Model.** Multimodal feature encoder can be BEVFusion [28] or UniTR [35]. More configuration details can be found in the previous paper [28, 35]. The number of perception queries is set to $N$=300. The multi-task decoder adopts $L$=3. Scene-level feature aggregation loops twice. For MWWA, the sub-BEV number $M$ is set to 4, 8 heads are used in the attention, and each 2 heads focus on the same scale. The window sizes $P_i$ are set to $3 \times 3$, $6 \times 6$, $9 \times 9$, and $18 \times 18$ respectively.

**Loss.** For the loss of 3D object detection $L_{3D}$, we adopt the focal loss [24] for classification and $L1$ loss for 3D bounding box regression, and the loss weights of the two are set to 2.0 and 0.25 respectively. We use the standard focal loss $L_{seg}$ for BEV map segmentation. The losses of the two tasks are simply added with weights to form the overall loss $L_{total} = \alpha L_{3D} + \beta L_{seg}$. To balance multiple training tasks, we set the weights $\alpha$ and $\beta$ to 3 and 1 respectively.

**Training.** Different from the training strategy of the general architecture [11, 28] that separates the BEV encoders for different tasks, our MaskBEV jointly trains 3D detection and BEV map segmentation tasks in the unified encoder-decoder framework. We trained our model on 8 NVIDIA A800 GPUs by AdamW optimizer [29]. We used a batch size of 8 and 24 and trained for 20 epochs for BEVFusion [28] and UniTR [35] encoder backbone. Both encoder backbone with the once-cycle learning policy [32] and a maximum learning

rate of 2e−3. We follow [28, 35] using the CBGS [53] strategy and the multi-modal data augmentation.

### 4.3 Main Results

Our MaskBEV is designed for MTL (3D object detection and BEV map segmentation), and we mainly focus on MTL methods on nuScenes. We use UniTR [35] as the feature encoding backbone network in experiments. As shown in Table 1, despite the negative transfer [28] of MTL, our MaskBEV achieves SOTA performance on MTL with 72.9 NDS and 73.9 mIoU and outperforms previous SOTA UniTR by +1.3 NDS and +2.7 mIoU. UniTR, like BEVFusion [28] and MetaBEV [11], adopts a separate BEV encoders strategy to perform MTL. MaskBEV outperforms MetaBEV by +3.1 NDS and +7.0 mIoU, where MetaBEV adopts an MTL optimization module. Furthermore, MaskBEV achieves comparable results to UniTR trained with single-task learning (STL) in 3D detection (72.9 NDS $v.s.$ 73.3 NDS) and map segmentation (73.9 mIoU $v.s.$ 74.7 mIoU). The results show that exploiting the complementary characteristics between multiple tasks improves the performance of MTL. Fig. 5 shows some qualitative results. The yellow and green marks show that our MaskBEV performance on MTL is far better than the UniTR performance on MTL, and is close to the UniTR performance on STL. See Fig. 6 for more visualizations and a video in Appendix.

### 4.4 Robustness Against BEV Encoder

To demonstrate robustness, we evaluate our MTL framework on different BEV feature encoder backbone networks [28, 35]. Since MetaBEV [11] is not open source, we do not use it as a baseline. All experiments are performed on the nuScenes val set. The results in Table 2 show that the performance of each method is impaired on MTL, but our MaskBEV can bring consistent improvements to them, which proves the effectiveness of MaskBEV on MTL. Specifically, for BEVFusion [28], our proposed multi-task head can obtain +1.2 NDS and +3.4 mIoU improvements compared to the separate BEV encoders strategy. The separation strategy proposed by BEVFusion can improve MTL performance. On the stronger baseline UniTR [35], our MaskBEV achieves improvements of +1.4 NDS and +2.7 mIoU. The results show that MaskBEV can be used as a general MTL framework.

Moreover, we compare inference latency with open source methods in Table 3. Taking BEVFusion [28] and UniTR [35] as baselines respectively, our methods both achieve slightly leading inference speed, but greatly improve the performance.

### 4.5 Ablation Studies

**Network configurations.** In Table 4, we analyze the impact of different segmentation mask thresholds. We observe that lower thresholds help increase mask coverage and improve performance, as we analyzed in Sec 3.3. Table 5 suggests that using only boxes as masks limits the performance. We aim to expand the potential regions of interest appropriately, and the 1.3 times enlargement of circular regions validates this idea. This brings a performance improvement of +0.6 mAP and validates our motivation.

**Scene-level feature aggregation.** In Table 6, we ablate the impact of scene-level feature aggregation. Both MWWA and ASPP positively contribute to the final performance. The $1^{st}$ and $2^{nd}$ rows

Table 1: Comparisons with previous state-of-the-art methods on nuScenes val set. 'L' and 'C' represent Lidar and camera, respectively. Single-task learning means the 3D detection and map segmentation are trained independently. Multi-task learning (MTL) means joint training of two tasks. ∗ indicates MTL results for a fair comparison. † represents object-centric methods, specifically designed for 3D detection, which are difficult to generalize to map segmentation. ‡ indicates separate BEV encoders. The best is in bold.

| Methods | Modality | mAP | NDS | Drivable | Ped.Cross | Walkway | StopLine | Carpark | Divider | Mean |
|---|---|---|---|---|---|---|---|---|---|---|
| Single-task learning | | | | | | | | | | |
| BEVFormer [22] | C | 41.6 | 51.7 | 80.1 | - | - | - | - | 25.7 | - |
| BEVFusion [28] | C | 35.6 | 41.2 | 81.7 | 54.8 | 58.4 | 47.4 | 50.7 | 46.4 | 56.6 |
| X-Align [2] | C | - | - | 82.4 | 55.6 | 59.3 | 49.6 | 53.8 | 47.4 | 58.0 |
| PETR v2† [26] | C | 42.1 | 52.4 | 85.6 | - | - | - | - | 49.0 | - |
| QAF2D† [16] | C | 50.0 | 58.6 | - | - | - | - | - | - | - |
| CenterPoint [42] | L | 59.6 | 66.8 | 75.6 | 48.4 | 57.5 | 36.5 | 31.7 | 41.9 | 48.6 |
| BEVFusion [28] | L | 64.7 | 69.3 | 75.6 | 48.4 | 57.5 | 36.4 | 31.7 | 41.9 | 48.6 |
| MetaBEV-T [11] | L | 64.2 | 69.3 | 87.9 | 63.4 | 71.6 | 55.0 | 55.1 | 55.7 | 64.8 |
| FocalFormer3D† [7] | L | 66.4 | 70.9 | - | - | - | - | - | - | - |
| SAFDNet† [45] | L | 66.3 | 71.0 | - | - | - | - | - | - | - |
| MVP [43] | L+C | 66.1 | 70.0 | 76.1 | 48.7 | 57.0 | 36.9 | 33.0 | 42.2 | 49.0 |
| TransFusion† [1] | L+C | 67.3 | 71.2 | - | - | - | - | - | - | - |
| BEVFusion [28] | L+C | 68.5 | 71.4 | 85.5 | 60.5 | 67.6 | 52.0 | 57.0 | 53.7 | 62.7 |
| X-Align [2] | L+C | - | - | 86.8 | 65.2 | 70.0 | 58.3 | 57.1 | 58.2 | 65.7 |
| MetaBEV-T [11] | L+C | 68.0 | 71.5 | 89.6 | 68.4 | 74.8 | 63.3 | 64.4 | 61.8 | 70.4 |
| MSMDFusion [17] | L+C | 69.3 | 72.1 | - | - | - | - | - | - | - |
| DeepInteraction† [40] | L+C | 69.9 | 72.6 | - | - | - | - | - | - | - |
| CMT† [38] | L+C | 70.3 | 72.9 | - | - | - | - | - | - | - |
| FocalFormer3D-F† [7] | L+C | 70.5 | 73.1 | - | - | - | - | - | - | - |
| SparseFusion† [52] | L+C | 71.0 | 73.1 | - | - | - | - | - | - | - |
| UniTR [35] | L+C | 70.5 | 73.3 | **90.5** | **73.8** | **79.1** | **68.0** | **72.7** | **64.0** | **74.7** |
| IS-FUSION† [41] | L+C | **72.8** | **74.0** | - | - | - | - | - | - | - |
| Multi-task learning | | | | | | | | | | |
| BEVFusion‡ [28] | L+C | 65.8 | 69.8 | 83.9 | 55.7 | 63.8 | 43.4 | 54.8 | 49.6 | 58.5 |
| MetaBEV‡ [11] | L+C | 65.4 | 69.8 | 88.5 | 64.9 | 71.8 | 56.7 | 61.1 | 58.2 | 66.9 |
| UniTR∗‡ [35] | L+C | 68.2 | 71.6 | 88.9 | 70.1 | 76.4 | 61.9 | 69.0 | 61.1 | 71.2 |
| **MaskBEV** | L+C | **69.8** | **72.9** | **90.0** | **73.1** | **78.4** | **66.8** | **71.9** | **63.1** | **73.9** |

Table 2: Comparison of the basic BEV feature encoding backbone networks on nuScenes val split.

| Method | Training Strategy | mAP | NDS | mIoU |
|---|---|---|---|---|
| BEVFusion [28] | STL | 68.5 | 71.4 | 62.7 |
| | MTL(shared) | - | 69.7 | 54.0 |
| | MTL(separate) | 65.8 | 69.8 | 58.5 |
| BEVFusion+MaskBEV | MTL(shared) | **67.3** | **71.0** | **61.9** |
| UniTR [35] | STL | 70.5 | 73.3 | 74.7 |
| | MTL(shared) | 67.6 | 71.4 | 69.5 |
| | MTL(separate) | 68.2 | 71.6 | 71.2 |
| UniTR+MaskBEV | MTL(shared) | **69.8** | **72.9** | **73.9** |

Table 3: Multi-task latency and performance on nuScenes val set. Latency is measured on an A800 GPU.

| Models | Latency (ms) | NDS | mIoU |
|---|---|---|---|
| BEVFusion [28] | 167.4 | 69.7 | 54.0 |
| BEVFusion+MaskBEV | **149.5** | **71.0** | **61.9** |
| UniTR [35] | 138.1 | 71.6 | 71.2 |
| UniTR+MaskBEV | **122.8** | **72.9** | **73.9** |

Table 4: Segmentation mask threshold.

| | mAP | NDS | mIoU |
|---|---|---|---|
| 0.1 | **69.8** | **72.9** | **73.9** |
| 0.2 | 69.7 | **72.9** | 73.8 |
| 0.4 | 69.5 | 72.7 | 73.3 |

show that extracting multi-scale features with windowed attention brings performance improvements to multi-task perception. Their complementary improvements in the $3^{rd}$ row are understandable since MWWA focuses on multi-granularity features and ASPP focuses on the global contexts.

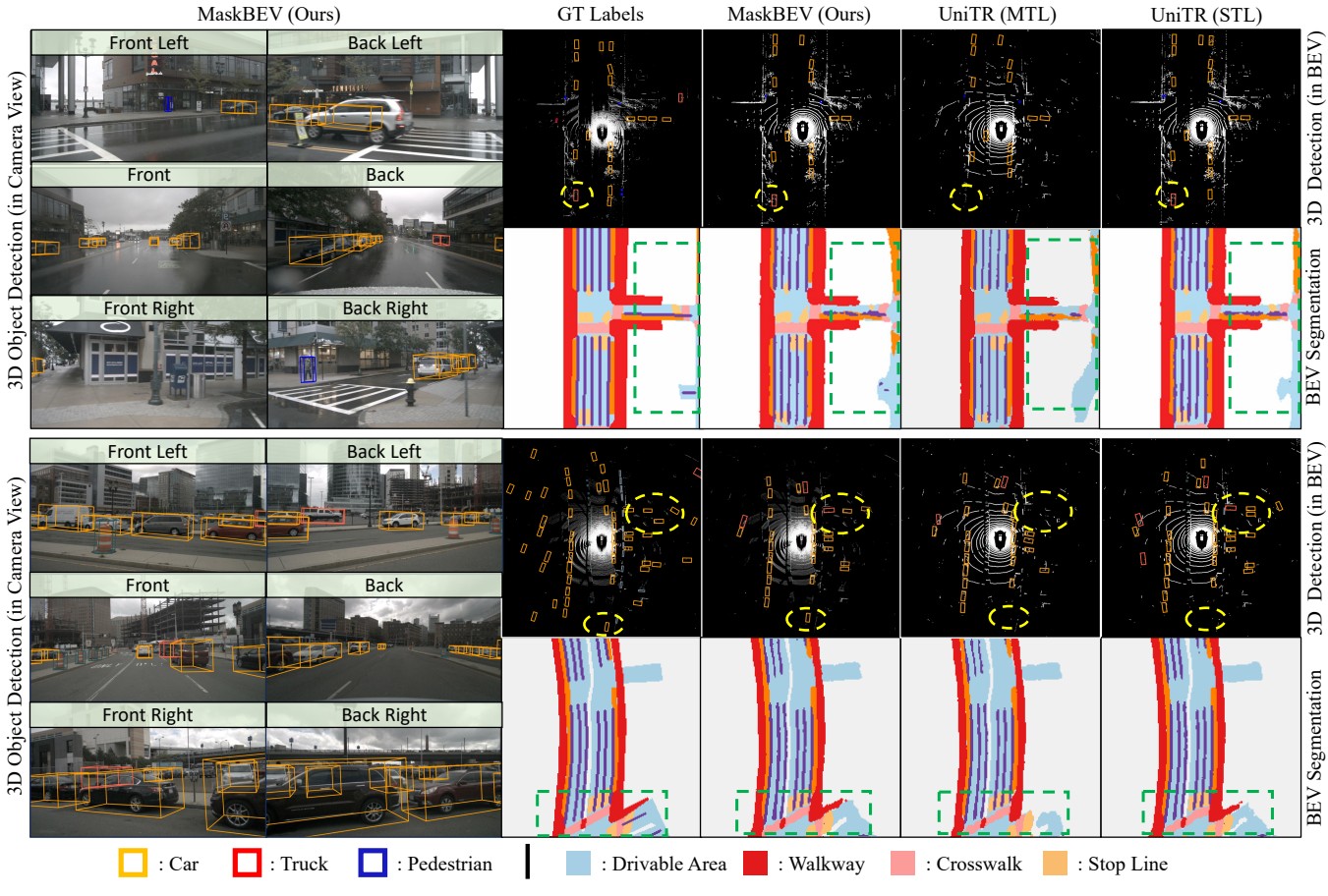

Figure 5: Qualitative results of MaskBEV on MTL, including 3D object detection and BEV map segmentation tasks. MaskBEV shows better results than the MTL variant of UniTR [35] and comparable results to the STL variant of UniTR.

Table 5: Detection mask design.

|        | mAP  | NDS  | mIoU |
|--------|------|------|------|
| Box    | 69.3 | 72.6 | 73.9 |
| Circle | 69.5 | 72.7 | **74.0** |
| 1.3 ×  | **69.8** | **72.9** | 73.9 |

Table 6: Ablation study on the scene-level feature aggregation

| MWWA | ASPP | mAP  | NDS  | mIoU |
|------|------|------|------|------|
| ✓    |      | 69.6 | 72.6 | 73.5 |
|      | ✓    | 69.3 | 72.5 | 73.3 |
| ✓    | ✓    | **69.8** | **72.9** | **73.9** |

## 4.6 Robustness Against Sensor Failure

We follow the same evaluation protocols adopted in UniTR [35] to demonstrate the robustness of our MaskBEV for Lidar and camera

Table 7: Robustness setting results of camera failure cases on nuScenes val set. F means the front camera. * indicates single-task training, and reported by UniTR [35]. UniTR† and our MaskBEV is multi-task learning.

| Method | Clean | | Missing F | | Preserve F | | Stuck | |
|--------|------|------|------|------|------|------|------|------|
|        | mAP  | NDS  | mAP  | NDS  | mAP  | NDS  | mAP  | NDS  |
| TransFusion* [1]  | 66.9 | 70.9 | 65.3 | 70.1 | 64.4 | 69.3 | 65.9 | 70.2 |
| BEVFusion* [23]   | 67.9 | 71.0 | 65.9 | 70.7 | 65.1 | 69.9 | 66.2 | 70.3 |
| UniTR* [35]       | **70.5** | **73.3** | **68.5** | **72.4** | **66.5** | **71.2** | **68.1** | **71.8** |
| UniTR† [35]       | 68.2 | 71.6 | 66.1 | 70.5 | 64.3 | 69.4 | 65.8 | 70.1 |
| UniTR+MaskBEV     | **69.8** | **72.9** | **67.9** | **71.9** | **66.0** | **70.6** | **67.6** | **71.5** |

malfunctioning. We refer readers to [23, 44] for more implementation details. As shown in Table 7 and 8, under certain camera and Lidar failure conditions, our MTL method shows comparable results with STL variant of UniTR, and outperforms MTL variant of UniTR. which proves the robustness of MaskBEV to sensor failure conditions.

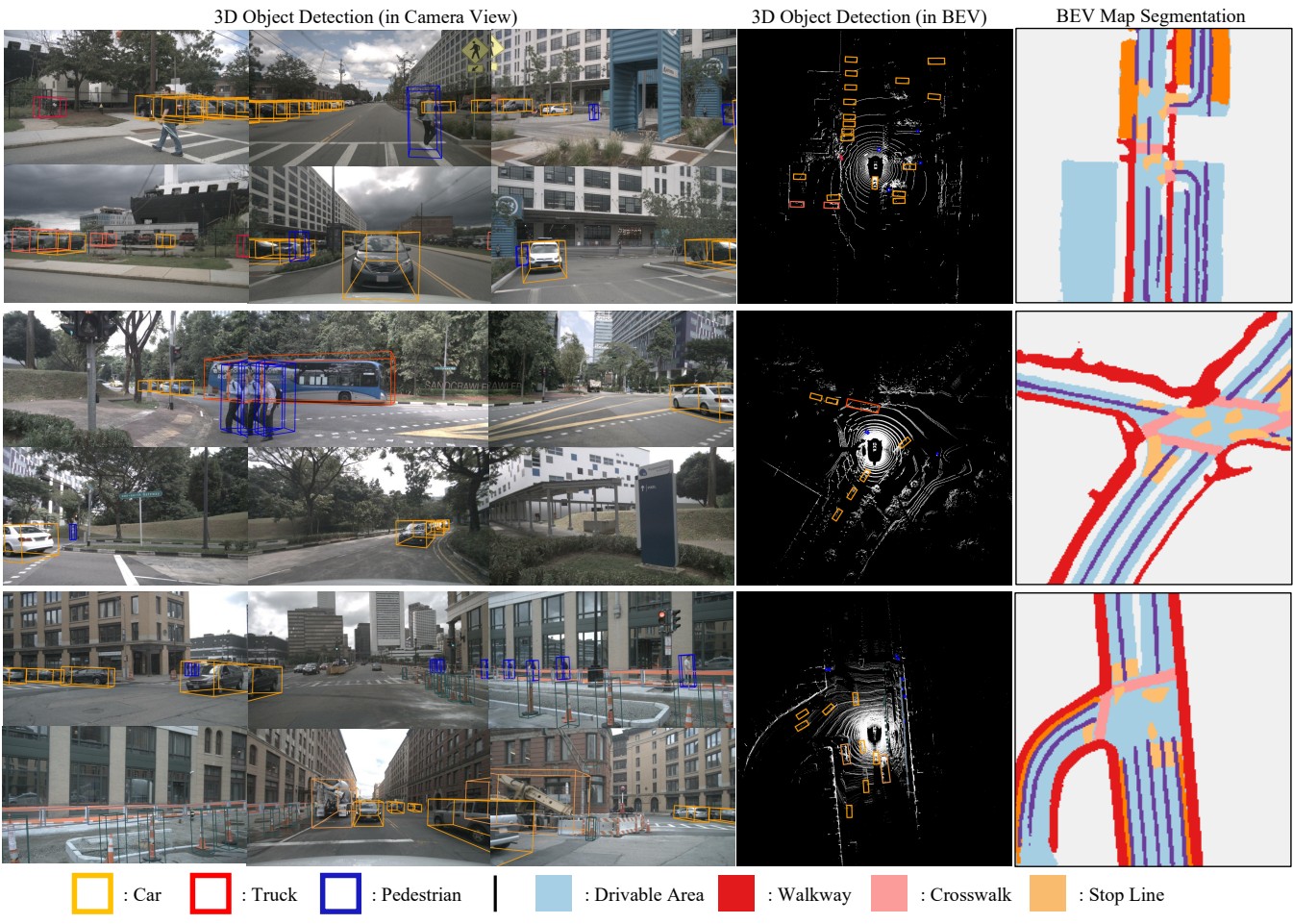

Figure 6: Qualitative results of MaskBEV on MTL, including 3D object detection and BEV map segmentation tasks.

Table 8: Ablation of Lidar beam failure with NDS evaluation metric. 'L' and 'C' represent Lidar and camera, respectively. † and our MaskBEV is multi-task learning.

| Method | C+L (1-beam) | C+L (4-beam) | C+L (16-beam) | C+L (32-beam) |
|---|---|---|---|---|
| BEVFusion [28] | 52.0 | 63.2 | 64.4 | 71.4 |
| MSMDFusion [17] | 45.7 | 59.3 | 69.3 | 72.1 |
| UniTR [35] | 59.5 | 68.5 | 72.2 | 73.3 |
| UniTR† [35] | 54.3 | 65.7 | 67.9 | 71.6 |
| MaskBEV | **57.3** | **67.8** | **71.4** | **72.9** |

## 5 CONCLUSION

This paper proposes a unified and general multimodal multi-task learning (MTL) paradigm. MaskBEV completes multi-task 3D perception based on bird's eye view (BEV) representation in a shared Transformer decoder. By fully exploiting the inherent dependencies between BEV map segmentation and 3D object detection tasks, MaskBEV alleviates the current performance degradation problem of MTL. MaskBEV breaks the common practice of designing specific decoding paradigms for specific perception tasks. MaskBEV achieves performance improvements and increased inference speed on MTL applications with multiple strong baseline methods. We believe that MaskBEV can provide a solid foundation for promoting the development of more efficient and universal MTL systems.

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
