# OpenReview forum: "MaskBEV: Towards A Unified Framework for BEV Detection and Map Segmentation"
_acmmm.org/ACMMM/2024/Conference — MM2024 Poster_

### Official Review · Reviewer_BVfn · 2024-05-24

**Rating:** 4
**Confidence:** 2

**Summary:**

The authors propose a task-agnostic Transformer decoder to process 3D object detection and BEV map segmentation via a unified decoder, which fully exploits the complementary information between BEV map segmentation and 3D object detection tasks in BEV space.

**Strengths:**

1) The author claims they are the first to unify transformer encoders to achieve simultaneous perception of 3D object detection and BEV map segmentation.
2) The proposed approach outperforms SOTA multi-task learning methods in both 3D object detection and BEV map segmentation tasks on nuScenes dataset.

**Limitations:**

I had two main concerns regarding their proposed unified transformer decoder below, I would like to increase the score if the authors can clearly explain it.
1) The architecture (Figure.2 ) of the unified transformer decoder in your paper is  the same as the one (Figure.2) in the previous publication Mask2Former [9], it is ok if the authors of both papers are the same, but it may break the double-blind review policy.
2) It is understood that multi-modal Camera and Lidar possess complementary information, the former more on color and texture, while the latter more on depth, but it is not clear how the complementary nature of the BEV map segmentation task and the 3D object detection task comes from? or just because your experimental results shows better results?

**Suitability:**

2

---

### Official Review · Reviewer_7Dxh · 2024-06-03

**Rating:** 4
**Confidence:** 2

**Summary:**

The paper introduces a novel multi-task learning framework, MaskBEV, designed to improve autonomous driving systems by unifying 3D object detection and BEV map segmentation through a single Transformer decoder. This approach utilizes advanced techniques like spatial modulation and scene-level feature aggregation to enhance performance. MaskBEV outperforms current state-of-the-art methods on the nuScenes dataset, demonstrating improvements in detection accuracy and segmentation.

**Strengths:**

1. Unlike existing methods that use separate task-specific decoders, MaskBEV employs a unified Transformer decoder.
2. The paper introduces technically sound methods such as spatial modulation to manage attention masks and scene-level feature aggregation to enhance feature representation. These techniques enhance overall system performance.
3. The evaluation of MaskBEV is thorough. The paper compares MaskBEV's performance against state-of-the-art MTL methods, demonstrating significant improvements in both 3D object detection and BEV map segmentation.

**Limitations:**

1. MaskBEV integrates 3D object detection and BEV map segmentation using a unified Transformer decoder, while may introduce complexity in terms of model architecture. A detailed analysis of the trade-offs between performance gains and computational overhead compared with existing MTL frameworks would be beneficial.
2. The integration of advanced techniques like spatial modulation and scene-level feature aggregation requires careful tuning of numerous parameters and design choices. This can make the optimization process more challenging.
3. The paper primarily evaluates the performance of MaskBEV on the nuScenes dataset. To further validate the effectiveness and robustness of the MaskBEV framework, it is recommended that the authors conduct experiments on additional datasets.
4. The paper demonstrates the effectiveness of MaskBEV for two specific tasks. However, its scalability to incorporate additional perception tasks (like motion prediction) is not discussed.

**Suitability:**

3

---

### Official Review · Reviewer_vyjj · 2024-06-04

**Rating:** 4
**Confidence:** 3

**Summary:**

This paper proposes a shared Transformer decoder like Mask2Former for unifying 3D object detection and bird’s eye view (BEV) map segmentation.

**Strengths:**

This paper proposes a shared Transformer decoder for unifying 3D object detection and bird’s eye view (BEV) map segmentation and achieves state-of-the-art performance on multi-task learning.

This paper proposes a spatial modulation strategy to obtain reliable multi-task masks and a scene-level feature aggregation module to capture multi-granularity and scene-level BEV contextual features.

**Limitations:**

1. The authors claim that the proposed framework fully exploits the complementary advantages between multiple tasks. It is intriguing to investigate whether the information from complementary multi-modal tasks outperforms single-modal tasks. The paper lacks a comparison between the single-task results of Maskbev and the multi-task results to illustrate this point.
2. The paper's ablation analysis is not comprehensive enough as it fails to include the overall ablation of the proposed spatial modulation strategy and feature aggregation module. For instance, it would be beneficial to compare the results of the baseline model against the baseline model with module 1, the baseline model with module 2, and the Maskbev model (baseline model with both module 1 and module 2).

**Suitability:**

3

---

### Official Review · Reviewer_ka4t · 2024-06-04

**Rating:** 4
**Confidence:** 3

**Summary:**

This paper introduces MaskBEV, a unified multi-task learning (MTL) framework for 3D object detection and bird’s eye view (BEV) map segmentation in autonomous driving. MaskBEV utilizes a masked attention-based Transformer decoder that processes both tasks simultaneously without requiring specific task heads. The proposed framework incorporates spatial modulation and scene-level context aggregation strategies to enhance performance. Experiments on the nuScenes dataset demonstrate that MaskBEV achieves significant improvements in both tasks compared to state-of-the-art MTL methods.

**Strengths:**

1. Extensive experiments on the nuScenes dataset demonstrate the effectiveness of MaskBEV.
2. Detection accuracy, sensor failures, and latency are evaluated, showing a clear overview of the method.

**Limitations:**

1. Based on my knowledge, most of current work use unified detetcion and segmentation frameworks. The claimed contribution of unified framework with shared encoder lacks novelty.
2. Missing citations:
[1] Planning-oriented autonomous driving. CVPR 2023
[2] Vad: Vectorized scene representation for efficient autonomous driving. ICCV 2023
[3] VLP: Vision Language Planning for Autonomous Driving. CVPR 2024
[4] Clip-bevformer: Enhancing multi-view image-based bev detector with ground truth flow. CVPR 2024

**Suitability:**

2

---

### Official Review · Reviewer_kXPd · 2024-06-08

**Rating:** 4
**Confidence:** 3

**Summary:**

This paper proposed MaskBEV, a novel method for unified multi-task learning in autonomous driving systems, focusing on adapting the fused camera and lidar features for 3D object detection and BEV map segmentation into a single Transformer-based framework. MaskBEV innovatively addresses the limitations of traditional approaches that handle detection and segmentation tasks separately by employing a spatial modulation strategy that capitalises on the integration of both tasks. This approach enhances how well the system understands and processes environmental data while also improves its efficiency. By testing this framework on the nuScenes dataset, MaskBEV demonstrates improvements in performance metrics and processing speed.

**Strengths:**

1. The MaskBEV framework demonstrated a novel approach using cross-modality features for multi-task learning which achieved state-of-the-art performance.
2. The evaluation displayed the efficiency of the proposed framework compared to other state-of-the-art methods in terms of computational complexity.
3. The ablation study showed the robustness of the framework against unexpected situations like sensor failures, where the method still maintained good performance.

**Limitations:**

1. While the method achieves state-of-the-art performance in multi-task learning approaches on the nuScenes dataset, it appears that all experiments were conducted only on this one dataset. Despite the size of the dataset and its variety of semantic settings, conducting more experiments on different datasets could further demonstrate the generalisation ability and performance of the model.
2. The evaluation demonstrated the efficiency of the approach by comparing the latency during processing. However, other measures, such as memory usage, could be take into account to further prove the advantages of the framework in terms of efficiency.
3. The ablation study section showed the robustness against sensor failure, this robustness could be further proven by testing under unexpected semantic settings such as heavy snow, or it could be tested on different camera settings such as lower resolution.
4. Despite claiming to mitigate effect of negative transfer in multi-task learning problems, the paper does not provide detailed evidence specifically showing how negative transfer is resolved compared to other methods.

**Suitability:**

3

---

### Official Review · Reviewer_65fz · 2024-06-08

**Rating:** 4
**Confidence:** 3

**Summary:**

The paper introduces MaskBEV, a novel masked attention multi-task learning (MTL) framework designed for 3D object detection on bird's eye view (BEV) map segmentation. MaskBEV integrates 3D object detection and BEV map segmentation into a single, task-agnostic Transformer decoder. Compared to previous works of separated BEV Encoders, this unified approach has better overall performance and learning efficiency by using the complementary nature of multi tasks. MaskBEV employs a masked attention mechanism, which selectively focuses on relevant regions in the BEV space, this improves computational efficiency and task accuracy. Also, spatial modulation and scene-level feature aggregation techniques are used to further enhance the model's ability to capture both geometric and semantic relationships in 3D object detection and BEV map segmentation.

The main contribution of this paper include its novel unified framework, utilizing multi-task complementary features. The framework begins by encoding camera and LiDAR data into feature tokens, which are then fused into a unified BEV feature map. A unified multi-task Transformer decoder, employing masked attention, processes these features to simultaneously generate 3D object detection and BEV map segmentation predictions, with scene-level feature aggregation, enhancing segmentation accuracy and making the approach theoretically sound and practically efficient.

**Strengths:**

1. Simplifies the architecture by using a single decoder for multiple tasks, reducing computational complexity and enhancing performance. Achieves state-of-the-art results in both 3D object detection and BEV map segmentation.
2. The use of masked attention is theoretically sound as it allows the model to selectively attend to relevant features, reducing noise and improving the efficiency of the attention mechanism. The paper provides a detailed explanation of the implementation of this mechanism, ensuring technical correctness.
3. Introduces spatial modulation and scene-level feature aggregation strategies to exploit complementary information between tasks. The theoretical basis for these methods is well-explained, and their integration on the MaskBEV framework is coherent and logical.
4. The clarity of the methodology ensures that the framework can be easily understood. The use of diagrams and detailed explanations helps in conveying complex concepts effectively.
5. The evaluation includes comparisons with several state-of-the-art methods, demonstrating MaskBEV's advantages in both 3D object detection and BEV map segmentation performance, providing strong empirical evidence for the efficacy of the proposed framework. The ablation experiment supports the importance of the masked attention mechanism, spatial modulation, and context aggregation.

**Limitations:**

1. English gramma issues. E.g. line 148 and 793.
2. Evaluating the inference speed on various hardware configurations (e.g., different types of GPUs) would help in understanding the practical deployment implications of MaskBEV.
3. The only autonomous driving dataset used in this paper is nuScenes. However, the paper would be benefit from comparing various datasets.
4. There might be specific scenarios, such as varying environment conditions, where separate BEV encoders could perform better. These scenarios are not thoroughly evaluated, and the potential advantages of separate encoders are not fully explored. Detailed performance metrics comparing unified and separate BEV encoders across different tasks and conditions would provide a more comprehensive understanding.
5. As shown in Table 7 and Table 8, the results of the experiments on both single-task learning(STL) and multi-task learning(MTL) indicates that the STL variant slightly outperforms MTL variant, there are still space to improve the model to outperform STL.
6. Evaluate and compare the performance metrics (e.g., NDS, mIoU) of MaskBEV with masked attention versus unmasked attention on the same dataset would be more comprehensive.

**Suitability:**

3

---

### Meta-Review · Area_Chair_xreR · 2024-06-26

**Recommendation:** Accept (Poster)
**Confidence:** 5

**Metareview:**

This paper was reviewed by six experts.

The initial ratings were consistently borderline accept, indicating that all reviewers' first impressions were generally supportive.

After the rebuttal, four reviewers finalized their ratings as borderline accept (one reviewer posted their final rating in the discussion channel).

The AC read the paper, the reviews, and the rebuttal, and would like to second all reviewers' "borderline accept" ratings.

---

### Meta-Review · Area_Chair_TKPk · 2024-07-06

**Recommendation:** Accept (Poster)
**Confidence:** 5

**Metareview:**

This paper presents a multi-task learning framework that unifies BEV map segmentation and 3D object detection. Particularly, the proposed framework utilizes a transformer-based architecture that first encodes and fuses images and lidar data in BEV space and then processes the corresponding tokens via a unified decoder. A spatial modulation strategy and a scene-level feature aggregation module are also introduced to better exploit contextual information. The proposed framework is evaluated on the nuScenes dataset and outperforms several existing methods.

All reviewers consistently rate this paper as “borderline accept” both before and after the rebuttal. While there are minor concerns such as typos and missing citations, the AC believes these issues can be addressed in the revised version. Therefore, the AC recommends accepting this paper.